Evidence from the resurrected family Polyrhabdinidae Kamm, 1922 (Apicomplexa: Gregarinomorpha) supports the epimerite, an attachment organelle, as a major eugregarine innovation

Paskerova Gita G. gitapasker@yahoo.com 1
Miroliubova Tatiana S. 2
Valigurová Andrea 3
Janouškovec Jan 4
Kováčiková Magdaléna 3
Diakin Andrei 3
Sokolova Yuliya Ya. 5
Mikhailov Kirill V. 6 7
Aleoshin Vladimir V. 6 7
Simdyanov Timur G. 8
1 Department of Invertebrate Zoology, Faculty of Biology, St Petersburg State University , St Petersburg , Russia
2 Laboratory for Fauna and Systematics of Parasites, Center for Parasitology, Severtsov Institute of Ecology and Evolution, Russian Academy of Sciences , Moscow , Russian Federation
3 Department of Botany and Zoology, Faculty of Science, Masaryk University , Brno , Czech Republic
4 Centre Algatech, Institute of Microbiology of the Czech Academy of Sciences , Třeboň , Czech Republic
5 Institute of Cytology, Russian Academy of Sciences , St Petersburg , Russian Federation
6 Belozersky Institute for Physico-Chemical Biology, Lomonosov Moscow State University , Moscow , Russian Federation
7 Kharkevich Institute for Information Transmission Problems, Russian Academy of Sciences , Moscow , Russian Federation
8 Department of Invertebrate Zoology, Faculty of Biology, Lomonosov Moscow State University , Moscow , Russian Federation
Gillespie Joseph
Electronic publication date: 2021 Sep 16
Publication date: 2021
Volume: 9
Electronic Location ID: e11912
Received 2020 Dec 16; Accepted 2021 Jul 14
Copyright: ©2021 Paskerova et al.
Copyright year: 2021
Copyright holder: Paskerova et al.
License: This is an open access article distributed under the terms of the Creative Commons Attribution License, which permits unrestricted use, distribution, reproduction and adaptation in any medium and for any purpose provided that it is properly attributed. For attribution, the original author(s), title, publication source (PeerJ) and either DOI or URL of the article must be cited.
License URL: https://creativecommons.org/licenses/by/4.0/

Keywords: Eugregarinida, Intestinal parasites, Marine gregarines, Ultrastructure, SSU and LSU rDNA, Host-parasite relationships, Environmental DNA sequences, Phylogeny, Taxonomy

Funding: The Russian Science Foundation project number 18-04-00123 The Russian Foundation for Basic Research 18-04-00324 18-04-01359 The Czech Science Foundation (project number GBP505/12/G112 (ECIP – Centre of excellence)) The phylogenetic analysis and scanning electron microscopy were supported by the Russian Science Foundation (project number 18-04-00123); fieldworks and transmission electron microscopy were supported by the Russian Foundation for Basic Research (grant numbers 18-04-00324, 18-04-01359) and the Czech Science Foundation (project number GBP505/12/G112 (ECIP – Centre of excellence)). The funders had no role in study design, data collection and analysis, decision to publish, or preparation of the manuscript.

==============================
Background

Gregarines are a major group of apicomplexan parasites of invertebrates. The gregarine classification is largely incomplete because it relies primarily on light microscopy, while electron microscopy and molecular data in the group are fragmentary and often do not overlap. A key characteristic in gregarine taxonomy is the structure and function of their attachment organelles (AOs). AOs have been commonly classified as “mucrons” or “epimerites” based on their association with other cellular traits such as septation. An alternative proposal focused on the AOs structure, functional role, and developmental fate has recently restricted the terms “mucron” to archigregarines and “epimerite” to eugregarines.

Methods

Light microscopy and scanning and transmission electron microscopy, molecular phylogenetic analyses of ribosomal RNA genes.

Results

We obtained the first data on fine morphology of aseptate eugregarines Polyrhabdina pygospionis and Polyrhabdina cf. spionis, the type species. We demonstrate that their AOs differ from the mucron in archigregarines and represent an epimerite structurally resembling that in other eugregarines examined using electron microscopy. We then used the concatenated ribosomal operon DNA sequences (SSU, 5.8S, and LSU rDNA) of P. pygospionis to explore the phylogeny of eugregarines with a resolution superior to SSU rDNA alone. The obtained phylogenies show that the Polyrhabdina clade represents an independent, deep-branching family in the Ancoroidea clade within eugregarines. Combined, these results lend strong support to the hypothesis that the epimerite is a synapomorphic innovation of eugregarines. Based on these findings, we resurrect the family Polyrhabdinidae Kamm, 1922 and erect and diagnose the family Trollidiidae fam. n. within the superfamily Ancoroidea Simdyanov et al., 2017. Additionally, we re-describe the characteristics of P. pygospionis, emend the diagnoses of the genus Polyrhabdina, the family Polyrhabdinidae, and the superfamily Ancoroidea.

Introduction

Apicomplexa are a large and diverse group of unicellular eukaryotes, many of which are symbionts of invertebrate and vertebrate animals (Simdyanov et al., 2018). Gregarines appear to be a monophyletic group of apicomplexan parasites—class Gregarinomorpha Grassé, 1953 (Janouškovec et al., 2019). They are characterised by the pre-sexual association of gamonts (syzygy) and gametocyst production (Simdyanov et al., 2017). At present, gregarines include two groups: archigregarines (order Archigregarinida Grassé, 1953) which possess characters of plesiomorphic state and develop in the intestine of polychaetes and sipunculids, and eugregarines (order Eugregarinida Léger, 1900) which are parasites of the tissue, intestine and other cavities of diverse invertebrates (Adl et al., 2019; Desportes & Schrével, 2013; Simdyanov et al., 2017). Traditionally, gregarines also contained neogregarines, now a clearly polyphyletic mixture of eugregarines (Cavalier-Smith, 2014; Simdyanov et al., 2017), and blastogregarines, which can be either considered as a separate class due to the lack of the gametocyst and syzygy (Simdyanov et al., 2018) or the sister group of archigregarines (Janouškovec et al., 2019). Eugregarines have been classified by the morphology of their dominant trophozoite and gamont stages into two groups. Septate eugregarines (Septata Lankester, 1885) have a fibrillar septum subdividing their cell into anterior and nucleated posterior parts and predominately parasitize terrestrial invertebrates. Aseptate eugregarines (Aseptata Chakravarty, 1960) lack the septum and inhabit mostly marine invertebrates (Desportes & Schrével, 2013; Simdyanov, 2007). Molecular phylogenies show that this classification is not natural because septate eugregarines are polyphyletic (Cavalier-Smith, 2014; Simdyanov et al., 2017).

Molecular phylogenetic studies of gregarines are limited by the availability of reference sequences and are largely based on small subunit (SSU) ribosomal DNA (rDNA) sequences (e.g., Diakin et al., 2016; Rueckert et al., 2018; Wakeman & Leander, 2013; Wakeman et al., 2017). Since many gregarine SSU rDNA sequences form long branches in molecular phylogenies, both archi- and eugregarines were thought to be polyphyletic and possess characteristics of a convergent origin (e.g., Cavalier-Smith, 2014). However, recent studies that used the concatenated sequences of SSU and large subunit (LSU) rDNA genes have recovered relationships within Apicomplexa with a resolution superior to the analyses inferred from SSU rDNA alone and revealed gregarines as monophyletic (Paskerova et al., 2018; Simdyanov, Diakin & Aleoshin, 2015; Simdyanov et al., 2017; Simdyanov et al., 2018). This conclusion was unambiguously supported by phylogenetic analyses based on 296 (Janouškovec et al., 2019) and 195 (Mathur, Wakeman & Keeling, 2021) concatenated protein-coding genes. The observation that concatenated rDNA sequences perform better at resolving deep phylogenetic relationships in a group than SSU rDNA alone is receiving increasing support in studies of protist diversity (Jamy et al., 2019).

Most gregarines are extracellular parasites which attach with an attachment organelle (AO) to one or several host cells. The AO is commonly classified as “mucron” or “epimerite” depending on whether the gregarine is aseptate or septate, respectively (Levine, 1971). Reassessing the organization, function, and developmental fate of the AO in gregarine microscopy literature and in the aseptate eugregarine Ancora sagittata (Leuckart, 1860) Labbé, 1899, Simdyanov et al. (2017) proposed to restrict the term “epimerite” to the AO in eugregarines and the term “mucron” to the AO in archigregarines. The epimerite is an anchoring organelle, originated de novo in front of the trophozoite anterior end, varying in size and shape, and usually lost in the gamont. The mucron represents the hypertrophied anterior end of the trophozoite and gamont and is usually small, rounded or sucker-shaped.

In this study, we test the hypothesis that the epimerite is a synapomorphic trait of eugregarines. For this, we (1) studied the AO structure in the aseptate eugregarine Polyrhabdina pygospionis Caullery et Mesnil, 1914 parasitizing the polychaete Pygospio elegans; (2) compared fine morphology of P. pygospionis to that of Polyrhabdina cf. spionis, the type species from Malacoceros fuliginosus; and (3) resolved phylogenetic relationships of P. pygospionis with other groups of aseptate gregarines within the order Eugregarinida, using the analyses of sequences of concatenated ribosomal operon genes (SSU, 5.8S, and LSU rDNA). Our analyses revealed a new deep branching clade of aseptate eugregarines with the epimerite-type AO. This clade resurrects the family Polyrhabdinidae Kamm, 1922 and belongs to the superfamily Ancoroidea Simdyanov et al., 2017.

Materials & Methods

Collection of polychaete hosts and isolation of gregarines

Bristle worms Pygospio elegans Claparède, 1863 (Spionidae, Polychaeta) were collected at two sites of the littoral zone near the Marine Biological Station of St Petersburg State University (Bolshoy Goreliy Island, Keret’ Archipelago, Chupa Inlet, Kandalaksha Bay, White Sea, 66°18.770′N; 33°37.715′E) and at the White Sea Biological Station of Lomonosov Moscow State University (Velikaja Salma strait, Kandalaksha Bay, White Sea, 66°33.200′N, 33°6.283′E) in the summer of 2002–2018. Polychaetes Malacoceros fuliginosus (Claparède, 1868) (Spionidae, Polychaeta) were collected under stones at the intertidal zone near the Roscoff Biological Station (English Channel, Atlantic Ocean, 48°43.652′N 3°59.285′W) in September 2010.

The examined animals were stored and dissected for isolation of parasites according to Paskerova et al. (2018). The released parasites or small fragments of the host intestine with attached gregarines were rinsed three times in seawater filtered through Millipore (0.22 µm), then fixed for electron microscopy.

Light microscopy

More than 100 polychaetes of P. elegans were investigated in squash preparations with living parasites (Figs. 1A, 1D–1H). Separate eugregarines isolated from the host intestines were also investigated in living preparations (Figs. 1B–1C, Video S1). The microscopes used for observation were Leica DM2500 equipped with DIC optics, Plan-Apo objective lenses, and DFC 295 digital camera (Leica, Germany); MBR-1 (LOMO, Russia) equipped with phase contrast and Canon EOS 300D digital camera; Zeiss Axio Imager.A1 equipped with phase contrast and DIC optics and Axio-Cam MRc5 digital camera (Carl Zeiss, Germany). Maximal dimensions of gregarine cells were measured with the ImageJ program (rsb.info.nih.gov/ij/); average (av) and standard deviation (SD) values were calculated (Table S1).

Figure 1 General morphology of the eugregarine Polyrhabdina pygospionis.

Light differential interference (A, C–H) and phase contrast (B) microscopy. All micrographs show gregarines with the anterior end facing up. (A–B) Gamonts without the epimerite, slightly compressed with the coverslip; the nucleus (N) has one nucleolus (n). (C) Slightly compressed gamont with the epimerite (ep). (D) Compressed gamont with the epimerite (ep). Note the collar (col) surrounding the epimerite base. Inset. Another compressed gamont with a seal-like (white arrow) structure under the pellicle (between black arrowheads) in the zone of separation of the epimerite (ep, heavily compressed) from the cell; scale bar is 10 µm. (E) Two small gamonts glued to the host cell debris. Note nuclei (N) with three (left) and one (right) visible nucleoli (n). (F) Young trophozoite being mechanically dislodged from the host cell. Note long tensile cords deriving from the destroyed epimerite (ep), the nucleus (N) with two nucleoli (n). (G–H). Epicytic crests (ec) of compressed gamonts: almost straight (G, adjacent crests are mostly parallel or in apposition to each other) and undulated (H, many areas where adjacent crests are in opposition to each other).

Electron microscopy

For scanning (SEM) and transmission (TEM) electron microscopy, small pieces of the polychaete intestine with attached eugregarines or free eugregarines released from the host gut lumen were fixed in 2.5% glutaraldehyde in 0.2 M cacodylate buffer (pH 7.4, final osmolarity 720 mOsm) for 2 h, washed in filtered seawater and postfixed in 2% osmium tetroxide in the same buffer for 2 h (Figs. 2A–2H; 3A–3D; 4A–4D). For visualization of the glycocalyx and other mucosubstances, the samples were additionally fixed with 3% glutaraldehyde-ruthenium red [0.15% (w/v) stock water solution] in 0.2M cacodylate buffer (pH 7.4) and post-fixed with 1% OsO4-ruthenium red in the same buffer (Figs. 2I–2J; 3E–3G). Fixation was performed at +4°C. Fixed samples were dehydrated in an ascending ethanol series. For SEM, the fixed and dehydrated samples were critical point dried in liquid CO2 and then coated with gold or platinum. The samples were investigated with GEMINI Zeiss Supra 40VP (Carl Zeiss, Germany) and JSM-7401F (JEOL, Japan) scanning electron microscopes. In total, more than 50 individuals of each species, P. pygospionis and P. cf. spionis, were examined by SEM. For TEM, samples of P. pygospionis, additionally dehydrated in an ethanol/acetone mixture and rinsed in pure acetone, were embedded in Epon-Araldite or Epon blocks. They were sectioned with ultramicrotomes Leica EM UC6 and Leica EM UC7 (Leica, Germany). Ultrathin sections were stained according to standard protocols and examined with LEO 910 (Carl Zeiss, Germany), JEM 2100 (JEOL, Japan), and JEM-1010 (JEOL, Japan) electron microscopes equipped with digital or film cameras. In total, more than 15 individuals of P. pygospionis were entirely sectioned and examined by TEM.

Figure 2 Fine structure of the eugregarine Polyrhabdina pygospionis.

Scanning (A–E) and transmission (F–J) electron microscopy. (A) General morphology of a mechanically dislodged trophozoite with the epimerite (ep) having a collar (col) at the base and longitudinal epicytic crests (ec) on the cell surface. (B–C) Epicytic crests of two trophozoites at high magnification: B, almost straight; C, undulated. (D–E) The anterior end (D) and posterior end (E) of gamonts. Note the area where epicytic crests terminate. (F) Cross-section in the middle of a gamont showing epicytic crests (ec), loops of the internal lamina (lp) under the bottom of grooves between the epicytic crests, and differentiation of the ectoplasm (ecto) from the rest of the cytoplasm with amylopectin granules (ag). (G–H) Oblique sections of epicytic crests (ec). Note loops of the internal lamina (lp), a micropore (mp) on the wall of the epicytic crest, and presumably excreted mucous material (m) between crests. (I–J) Transversal section of finger-like epicytic crests (ec); J is a detail of I at a higher magnification. Note the three-membrane pellicle consisting of the plasma membrane (pm) covered by glycocalyx and the inner membrane complex (IMC) underlain by the internal lamina (il). In the apex of each crest, there are 10–12 apical rippled dense structures (aa) between the plasma membrane and IMC, and 10–12 apical filaments (af) under the IMC.

Figure 3 Attachment organelle (epimerite) of the eugregarine Polyrhabdina pygospionis.

Scanning (A–C) and transmission (D–G) electron microscopy A. Trophozoite (p) embedded in a piece of the ciliated intestinal epithelium of the host (h). (B) Trophozoite (detail of Fig. 2A at a higher magnification) having the epimerite (ep) with a damaged apical surface membrane (am). Note longitudinal epicytic crests (ec) starting from under the collar (col) of the epimerite base. (C) Micrograph showing a broken off epimerite embedded in the host intestinal epithelium (h). Note epicytic crests (ec) and the collar (col) at the epimerite base. (D) Longitudinal section of a trophozoite (p), attached to the host cell (h) with the epimerite (ep), showing the differentiation of the peripheral cytoplasm into ectoplasm (ecto) poor in amylopectin granules (ag), epicytic crests (ec) starting from under the epimerite collar (col), internal lamina (il) under the gregarine pellicle, granular material (grm) of the ectoplasm at the epimerite base. Note the cortical zone (cz) of the epimerite filled with finely granular material. The infected host cell has vesicles with the material of heterogeneous electron density (vdm). Insert. A close up of part D at a higher magnification shows the terminal sections of the internal lamina (il) and the inner membrane complex (IMC), the narrow and deep circular gap (gp) underlain by a fibrillar layer (f), the distal end of the circular host cell fold (hf) covering the collar and embedded in the gap. Note granular material (grm) with short fibrils mainly congregated near the termini of the internal lamina and IMC and a kind of tight cell junction between plasma membranes of the parasite AO (pm) and the host epithelial cell (hm). (E) Longitudinal section of an epimerite embedded into an invagination of the host cell (h). Note the collar (col) and granular material (grm) at the epimerite base, fine electron-dense granular material in the cortical zone (cz) and the cytoplasm filled with amylopectin granules (ag) in the middle of the epimerite. (F) Details of the epimerite collar (col) base. The cortical zone (cz) of the collar has fine electron-dense granular cytoplasm and is covered by the parasite plasma membrane (pm) and a thin circular host cell fold (hf), the distal end of which was embedded in the circular gap (gp) underlain by the fibrillar layer (f). Note the granular material (grm) and the termini of the internal lamina (il) and inner membrane complex (IMC). (G) Contact zone between the host (h) and parasite (p) cells. Note the host (hm) and parasite (pm) plasma membranes, and fine electron-dense granular material in the epimerite cortical zone (cz).

Polyrhabdina rRNAs assembly

The rRNA of P. pygospionis was assembled from transcriptomic data generated from about 19 parasite cells isolated from five P. elegans polychaetes which were collected at Velikaya Salma strait, Kandalaksha Bay, White Sea, in 2016 (Janouškovec et al., 2019). Assembled, high-coverage transcripts corresponding to P. pygospionis ribosomal RNA were identified by BLASTN homology searches and joined at overlaps into eight larger contigs. Raw transcriptomic reads mapping onto the eight rRNA contigs were retrieved in Bowtie 2 (default settings) and used for extension of the contigs in Consed v29 with the following crossmatch parameters: -minmatch 50 -minscore 50 -penalty -9. Repeating this process in several iterations allowed us to merge and subsequently validate the complete rRNA operon sequence of P. pygospionis (comprising the SSU, ITS1, 5.8S, ITS2, and LSU). The final sequence was deposited into GenBank under the accession number MT214481.

We did not isolate DNA from P. cf. spionis cells because we found a limited number of cells, mostly of which were fixed for electron microscopy (see Results).

Figure 4 General morphology of the eugregarine Polyrhabdina cf. spionis.

Scanning electron microscopy (A). Several trophozoites (p) attached to the host intestinal epithelium (h). (B) Attached trophozoite (p) with the epimerite (ep) partly embedded into the host intestinal epithelium (h) and longitudinal epicytic crests (ec). (C) Details of a detached trophozoite with the preserved epimerite. Note that longitudinal epicytic crests (ec) emerge from under the collar (col) at the base of the globular epimerite (ep). (D) Details of a mechanically dislodged trophozoite. The apical part of the epimerite (ep) embedded in the host cell was broken off, while the collar (col) at the epimerite base was retained.

Molecular phylogenetic analysis

The rDNA dataset for phylogenetic analyses was constructed using publicly available sequence data and designed to maximize the diversity of eugregarines at the family level. Unidentified environmental sequences and rDNA contigs from metagenome assemblies were found by BLAST searches (Altschul et al., 1997) in the nr and wgs databases of NCBI. Small set of Coccidiomorpha (Coccidia and Hematozoa) species was used as an outgroup in the phylogenetic analyses.

Three datasets were prepared for phylogenetic analyses: taxonomically balanced SSU rDNA dataset (94 OTUs), a dataset without divergent long-branch OTUs (65 OTUs), and a concatenated dataset with SSU, 5.8S, and LSU rDNA sequences (31 OTUs). The datasets were aligned in MUSCLE 3.6 (Edgar, 2004) and manually adjusted with BioEdit 7.0.9.0 (Hall, 1999): gaps, columns containing few nucleotides or hypervariable regions (V2, V4, V7, and V9) were removed, which resulted in 1,574-site (SSU) and 4,571-site (concatenated) alignments. To verify the results of manual masking, we additionally used two different automatic alignment and masking strategies for the SSU datasets. With the first strategy, the alignment of SSU rDNA sequences was done with MAFFT (Katoh & Standley, 2013) using a combination of local and structural alignments: the initial draft alignment was prepared using the local alignment with generalized affine gap cost (E-INS-i), the variable gap-rich regions were then aligned individually with the secondary structure aware alignment (X-INS-i) employing MXSCARNA (Tabei et al., 2008) to produce the final alignment. Prior to analyses the alignment was masked with trimAl (Capella-Gutierrez, Silla-Martinez & Gabaldon, 2009) using a gap threshold of 0.5 and a minimum block size of 3; the result was a 1,578-site alignment. The second strategy employed the GUIDANCE2 web server (Landan & Graur, 2008; Penn et al., 2010; Sela et al., 2015) and MAFFT (E-INS-i) to generate a series of alignments with varying confidence score thresholds. Columns with confidence scores below 0.715, 0.794, 0.900, 0.942, 0.970, 0.973, and 0.990 were removed resulting in 1,574, 1,471, 1,366, 1,257, 1,126, 1,087 and 828-site long alignments, respectively. To obtain a similar series for the alignment generated using the first strategy, we calculated site-specific evolutionary rates with IQ-TREE 2.1.2 (Minh et al., 2020) and preformed stepwise removal of sites starting with the fastest category.

Maximum-likelihood (ML) analyses were performed with IQ-TREE 2.1.2 (Minh et al., 2020) using non-parametric bootstrap (-b 1000) and ultrafast bootstrap approximation (UFBoot, bb 1000) (Minh, Nguyen & von Haeseler, 2013) employing the CIPRES Science Gateway (Miller, Pfeiffer & Schwartz, 2010). Bayesian inference (BI) analyses were done with MrBayes 3.2.6 (Ronquist et al., 2012), PhyloBayes (Lartillot et al., 2013), and Phycas 2.2 (Lewis, Holder & Swofford, 2015). Evolutionary models for ML and Bayesian analyses were selected with ModelFinder (Kalyaanamoorthy et al., 2017): the GTR+F+I+G8 model was selected for the SSU rDNA datasets, and the SSU and LSU partitions in the concatenated dataset, while the GTR+F+G8 model was selected for the 5.8S partition. The following parameters of Metropolis Coupled Markov Chain Monte Carlo (mcmcmc) were used: nchains = 8, nruns = 2, temp = 0.025, ngen = 5,000,000, samplefreq = 1,000, burninfrac = 0.5. The average standard deviations of split frequencies at the end of BI (MrBayes) computations were 0.005044 for the 94 OTU dataset, 0.012636 for the 65 OTU dataset, and 0.003799 for the 31 OTU dataset. The ML support values were assigned to the SSU Bayesian tree using 1000 non-parametric bootstrap trees from the ML analysis via the –sup option of IQ-TREE 2.1.2 (Minh et al., 2020). Additional BI analysis of full manually masked SSU alignment (1,574 bp) was performed with PhyloBayes (Lartillot et al., 2013) under the GTR+ G8+CAT model (four chains, 20,000 cycles, the first half of sample points were discarded as burn-in). Constrained tree search and approximately unbiased (AU) tests were performed with IQ-TREE 2.1.2 (Minh et al., 2020).

Differences in the ML trees obtained with X-INS-i, GUIDANCE2 or full manually masked alignments were outlined by means of the principal component analysis (PCA). Using the bpcomp program of PhyloBayes, the ML trees were represented as sets of bipartitions with the corresponding bipartition support values treated as variables for PCA. For the analysis we used the support values calculated with UFBoot by IQ-TREE. The set of tree bipartitions used for PCA was reduced from the initial full set to just 146 bipartitions, representing only the interrelationships of main gregarine groups—groups that were well-supported by all analyses. The PCA was performed using R (R Core Team, 2021) and R packages FactoMineR (Lê, Josse & Husson, 2008), factoextra, and ggplot2 (Wickham, 2016).

The secondary structure of helix 17 (numbering according to Wuyts, Van de Peer & De Wachter, 2001) of SSU rRNA was predicted using the Mfold server (Zuker, 2003) at http://www.unafold.org/mfold/applications/rna-folding-form.php.

New zoological taxonomic names

The electronic version of this article in Portable Document Format (PDF) will represent a published work according to the International Commission on Zoological Nomenclature (ICZN), and hence the new names contained in the electronic version are effectively published under that Code from the electronic edition alone. This published work and the nomenclatural acts it contains have been registered in ZooBank, the online registration system for the ICZN. The ZooBank LSIDs (Life Science Identifiers) can be resolved and the associated information viewed through any standard web browser by appending the LSID to the prefix http://zoobank.org/. The LSID for this publication is: urn:lsid:zoobank.org:pub:693369E6-B319-4BB1-8E61-148FC4F5B271. The online version of this work is archived and available from the following digital repositories: PeerJ, PubMed Central SCIE and CLOCKSS.

Results

Polyrhabdina pygospionis

Occurrence

Eugregarines P. pygospionis were found in the intestine of 126 out of 302 (42%) examined Pygospio elegans polychaetes (Spionidae). The intensity of infection usually varied from 1 to 50 (mode 1, average (av.) 6) gregarines per host. Parasites P. pygospionis co-occurred with other symbionts, archigregarines Selenidium pygospionis Paskerova et al., 2018, in the gut of 66 polychaetes (52%) versus 60 worms (48%) infected only by eugregarines. From the life cycle stages, trophozoites (attached eugregarines) and gamonts (non-attached eugregarines) were found in the host intestine.

General and fine structure

The cell shape of gamonts varied from ellipsoid sometimes slightly curved to pear-shaped. Gamonts were circular in cross section, with rounded anterior and posterior ends. No septum was observed (Figs. 1A–1F, 2A, 2F; Table S1). Gamonts had a large, oval or almost spherical nucleus positioned longitudinally in the widest part of the cell, usually closer to the anterior end. In large gamonts, one large spherical nucleolus was observed in the nucleus (Figs. 1A–1B). In small gamonts, two to four spherical nucleoli of diverse sizes were situated at opposing nuclear poles (Figs. 1E–1F). Gamonts moved by gliding without obvious changes in the cell shape (Video S1).

The cell surface of gamonts had numerous longitudinal epicytic crests (or folds; in this study, we use the terminology of Simdyanov et al. (2017)). They were almost straight (Figs. 1G; 2A–2B) or undulated (Figs. 1H; 2C). At the parasite ends, epicytic crests passed into smooth areas, usually smaller at the anterior end and larger at the posterior one (Figs. 2D–2E). The cortex organization of the finger-like crests was typical of eugregarines: a three-membrane pellicle of 32–36 nm thick consisting of the plasma membrane and inner membrane complex (IMC). The pellicle was covered by a cell coat (glycocalyx) of about 10 nm and underlain by the internal lamina, an electron-dense fibrillar layer (Figs. 2F–2J). The internal lamina of 17–23 nm thick formed additional loops under the bottom of grooves between the epicytic crests. No links in the base of each crest were visible, due to which the crest cytoplasm communicates freely with the bulk of the cell cytoplasm (Figs. 2G, 2I). In the crest tip, there were 10–12 apical rippled dense structures (apical arcs) between the plasma membrane and the IMC as well as 10–12 apical 12-nm filaments under the IMC (Fig. 2J). Micropores typical for apicomplexans were present on lateral walls of epicytic crests (Figs. 2G–2H). Electron-dense material, presumably excreted mucus, was observed in between crests in some epicyte regions of parasite cells (Fig. 2G). In Polyrhabdina cells, the ectoplasm, a peripheral cytoplasm layer free of amylopectin, and the endoplasm, the bulk cytoplasm enriched with rounded amylopectin granules, were not distinctly separated (Fig. 2F). The thickness of the ectoplasm varied from 0.2 to 0.6 µm in the middle cell region; anterior and posterior zones of ectoplasm were usually equally sized.

The trophozoites were anchored in the host intestinal epithelium by a dome-shaped AO (Figs. 3A–3C; Table S1). Rarely, this organelle was also observed in gamonts (Figs. 1C–1D). The AO in trophozoites was almost entirely embedded into a deep invagination of the host epithelial cell. A circular fold facing posteriorly—the collar—was presented at the base of the AO (Fig. 1D). This collar was located above the apical surface of the host cell and limited the area of AO insertion into the host cell invagination (Figs. 3B–3D). Epicytic crests started from the AO base under the collar (Fig. 3C). Trophozoites mechanically dislodged from the intestinal epithelium during material preparation had an AO with a damaged apical surface corresponding to the host-parasite attachment site (Fig. 3B). The intact AO was covered only by the parasite plasma membrane, and the IMC terminated at the AO base (Figs. 3D–3G). The attachment site had an appearance of a tight cell junction without a distinct gap between plasma membranes of the AO and host cell. Both membranes were underlain by electron-dense areas of fibrillar-like appearance (Figs. 3D–3G). The collar represented a thin extension of the AO and was covered by the parasite plasma membrane (Figs. 3D–3F 3D, 3F). The cytoplasm of the AO was distinctly differentiated into two zones: a finely granular cortical zone located peripherally under the plasma membrane and within the collar, and a vesicular zone positioned in the centre and having occasional organelles typical for the rest of the parasite body, e.g., amylopectin granules (Figs. 3D, 3E). The infected host cell formed a thin circular fold at its apical surface. This fold consisted of two closely adjacent plasma membranes almost without any cytoplasm in between them and tightly surrounded the epimerite collar of the attached parasite (Fig. 3D). The distal end of the host circular fold was embedded in a gap of about 45 nm width and 600 nm depth located circularly at the base of the parasite AO under the collar. In the parasite cytoplasm, this circular gap was underlain by a fibrillar layer - the endpoint of the IMC. The terminus of the internal lamina was located under this layer (Figs. 3D inset, 3F). The ectoplasm of this region contained granular material with short fibrils mainly congregated near the termini of the internal lamina and IMC (Figs. 3D–3F).

According to our observation on living P. pygospionis cells, during mechanical AO separation from the parasite cell, the edges of the anterior end seem to be pulled together and sometimes only a small part of the cytoplasm escaped (Video S1). The wound surface was probably sealed by the granular cytoplasm with short fibrils located in the AO base (Fig. 1D inset; Fig. S1A). The fine structure of the anterior end of recently detached gamonts corresponded to our in vivo observations (Fig. S1B). In some parasites mechanically separated from the host cell by the pressure of a coverslip, long tensile cords were derived from the epimerite cytoplasm and the underlying ectoplasm (Fig. 1F). Basing on these obtained data, we presume that P. pygospionis gamonts usually discard their epimerites and, as a result, detach from the host tissue.

All parasitized host cells had an altered appearance: near the host-parasite contact zone, they had electron-dense, organelle-depleted cytoplasm with vesicles containing material of heterogeneous electron density, sometimes surrounded by an additional membrane (Fig. 3D). This may be considered as a host response to the gregarine infection.

Some examined eugregarines P. pygospionis were infected with microsporidia Metchnikovella incurvata Caullery and Mesnil, 1914 and M. spiralis Sokolova et al., 2014 (Figures S1A–S1B). Interestingly, these microsporidia also occupied the AO of infected gregarines (not shown).

Polyrhabdina cf. spionis (Von Kölliker, 1845) Mingazzini, 1891

Occurrence

Eugregarines P. cf. spionis were found in the intestine of 3 out of 18 (17%) Malacoceros fuliginosus polychaetes (Spionidae). One worm was relatively heavily infected, about 50 parasites per host, while others had only several eugregarines. From the life cycle stages, mainly trophozoites were found in the host intestine.

General morphology

Trophozoites were spindle- or rhomboid-shaped, wide in the cell middle and with a rounded posterior end (Fig. 4A; Table S1). A single rounded nucleus was in the widest part of the cell (not shown). Trophozoites were anchored in the intestinal epithelium by a globular AO with the collar around its base (Figs. 4B–4D). In some individuals, the collar was located above the apical surface of the host cell (Fig. 4A). The cell surface of parasites was covered with longitudinal epicytic crests, straight or slightly undulated. Epicytic crests started from the AO base under the collar (Fig. 4B). The parasites moved by gliding (not shown).

Figure 5 Bayesian tree of eugregarines inferred from the manually masked dataset of 94 SSU rDNA sequences and 1,574 sites under the GTR+F+I+G8 model.

Numbers at branches indicate Bayesian posterior probabilities (numerator) and ML bootstrap percentage (denominator). Black dots on the branches indicate Bayesian posterior probabilities and bootstrap percentages of 1.0 and 100%, respectively. The newly obtained sequence of Polyrhabdina pygospionis is in bold. The names of major eugregarine lineages correspond to Simdyanov et al., 2017 and Cavalier-Smith, 2014.

Phylogenies inferred from rDNA sequences

SSU rDNA phylogenies inferred by the Bayesian inference (BI) in MrBayes (Fig. 5) and Phycas and by the Maximum-likelihood (ML) analyses (not shown) using the full manually masked alignment showed almost identical topologies with a few minor differences. In both phylogenies, the sequence of Polyrhabdina pygospionis was placed in a robustly supported clade of 23 environmental sequences derived from marine sediments and one sequence derived from the foraminiferan Ammonia beccarii (see Discussion). This clade is subdivided into two subclades and is grouped with a clade consisting of Trollidium akkeshiense and five environmental sequences with high posterior probability (PP = 1.0) but low non-parametric bootstrap percentage (BP = 63%) supports. A clade containing Ancora sagittata and Polyplicarium species appears sister to the group with Polyrhabdina and Trollidium with statistically significant PP = 0.98, but low BP = 40%. BI analysis performed in PhyloBayes shows the same composition of Ancoroidea, but alternative topology with rearrangement of subclades within Polyrhabdinidae (Fig. S2). The analyses of the concatenated dataset (SSU rDNA+5.8S rDNA+ LSU rDNA) improved the resolution of the tree (Fig. 6), however, the taxonomic sampling was reduced and the phylogenetic diversity of gregarines was lower (see Materials and Methods). The position of P. pygospionis did not change in this tree: it was still a sister taxon to Trollidium with moderate support (PP = 0.95, BP = 79%). Importantly, the Trollidium-Polyrhabdina and Ancora dichotomy received stronger support as well (PP = 1, BP = 90%). Judging by the result of BI and the ML analyses, Polyrhabdina and relatives are presumably members of the superfamily Ancoroidea (see Discussion).

Figure 6 Bayesian inference tree of eugregarines obtained from the manually masked dataset of 31 concatenated SSU, 5.8S, and LSU rDNA sequences (4,571 sites).

Designations are the same as in Fig. 5.

The Ancoroidea clade uniting Polyrhabdina pygospionis, Trollidium akkeshiense, Ancora sagittata, Polyplicarium spp., and related environmental sequences is consistently present in the ML trees based on the dataset constructed using structural alignment (X-INS-i) as well as with the derivative alignments after sequential deletion of rapidly evolving sites, supporting the results obtained with manually masked alignments (Fig. S3). However, these phylogenies are generally in disagreement with the results obtained with the GUIDANCE2 alignments, which were generated with varying confidence score thresholds (Fig. S3). The GUIDANCE2 trees show less reconciliation in the topology and composition of subclades for the Ancoroidea, which usually involve the clade of Trichotokara and Paralecudina, but not necessarily Trollidium. Close relationship of the Trichotokara + Paralecudina clade with Ancoroidea is also observed with the manually masked dataset if CAT model of molecular evolution implemented in PhyloBayes program is applied (Fig. S3) or the most divergent sequences are excluded (65 OTUs) (Fig. S4). In an attempt to resolve the discrepancy between the results of structural alignment and GUIDANCE2 experiments, we tested the possible composition of the Ancoroidea. We built constrained trees with four different subclade sets (including or excluding Trollidium, Trichotokara and Paralecudina, and Cephaloidophora) for full and 1471-site alignments generated with GUIDANCE2 and structural alignment, and performed AU tests (Table S2). The AU tests did not reject the tested alternatives at the 5% significance level, although the p-values were highly dependent on the alignment used. We also performed the PCA for resolving main differences in topologies generated using three different alignment and masking strategies (Fig. S5). These differences were in Ancoroidea composition and Trichotokara + Paralecudina clade position, and some other systematic differences with a smaller contribution for these series of topologies. However, the PCA showed a preference for manual strategies as the obtained trees were more stable (red dots formed a denser group than green or blue ones in the data matrix, Fig. S5).

A shared feature of most Polyrhabdinidae, Trollidium, and related environmental sequences was found in the predicted structure of helix 17 of the SSU rRNA (numbering according Wuyts, Van de Peer & De Wachter, 2001). In most of these OTUs, the 3′-strand of helix 17 contains an additional nucleotide, forming a second bulge in the helix, while the typical state for eukaryotes is a single 1-nucleotide bulge in the 3′-strand (Fig. S6). No similar structures of helix 17 have been found outside Polyrhabdinidae, Trollidium, and related environmental sequences, however, within the group the helix 17 is not constant and evolves along the phylogenetic tree. This feature is an additional evidence for Polyrhabdina-Trollidium relationships as the additional bulge in helix 17 was not included in any of the alignments used in this research.

Discussion

Polyrhabdina species and their taxonomic position

The genus Polyrhabdina (original spelling Polyrabdina) was established by Mingazzini (1891) with Gregarina spionis Kölliker, 1848, a gregarine isolated from the polychaetes Scolelepis fuliginosa (Claparède, 1868) (now Malacoceros fuliginosa), as the type species. This genus was assigned to the family Lecudinidae Kamm, 1922 (Clopton, 2000; Ganapati, 1946; Grassé, 1953; Levine, 1971; Levine, 1977; Reichenow, 1929; Rueckert et al., 2018) or the family Polyrhabdinidae Kamm, 1922 (Desportes & Schrével, 2013; Kamm, 1922) depending on the emphasis given to gregarine septation and, correspondingly, AO naming—mucron or epimerite.

To date, the genus Polyrhabdina includes seven named species. One species has no description—Polyrhabdina sp. from Dipolydora socialis. All known Polyrhabdina spp. occur in the intestine of spionid polychaetes. All these species have been studied only using light and, in some cases, scanning electron microscopy (Caullery & Mesnil, 1897a; Caullery & Mesnil, 1897b; Caullery & Mesnil, 1914a; Caullery & Mesnil, 1919; Von Kölliker, 1845; Von Kölliker, 1848; Mingazzini, 1891; Reichenow, 1932; Mackinnon & Ray, 1931; Ganapati, 1946; Kamm, 1922; Fowell, 1936; De Faria, De Cunha & Da Fonseca, 1918; Rueckert et al., 2018; Table S3). No molecular sequence data are available for Polyrhabdina except for two sequences published in Rueckert et al. (2018), which are currently retracted from GenBank to clarify potential fungal contamination (Dr. Rueckert, pers. comm., 2020).

In Polyrhabdina spp., the only life cycle stages that have ever been described are the trophozoite, usually with a globular AO embedded in the host epithelium, and the gamont that usually lost its AO during separation from the host tissue. In trophozoites, some authors observed hook-like processes on the apical AO surface in addition to a circlet of tiny prongs (“teeth”) at its base (Caullery & Mesnil, 1914a; Ganapati, 1946; Von Kölliker, 1848; Mackinnon & Ray, 1931; Reichenow, 1932; Rueckert et al., 2018). In contrast, others described the AO with a collar, a posteriorly oriented circular fold, at its base and sometimes with a ring of small prongs, anterior to the collar (Fowell, 1936; Léger, 1893; Mackinnon & Ray, 1931). There was no consensus in these works on calling the AO a mucron or epimerite (Table S3).

Caullery and Mesnil recorded the eugregarine Polyrhabdina pygospionis in polychaetes Pygospio seticornis (now P. elegans) from the English Channel but did not provide an adequate description of this species. The host name and the infection caused by microsporidia Metchnikovella incurvata and M. oviformis are available evidence from this species (Caullery & Mesnil, 1914a; Caullery & Mesnil, 1914b; Caullery & Mesnil, 1919). We believe that we collected namely eugregarines P. pygospionis in the White Sea because they parasitized P. elegans polychaetes and were found to be infected by the microsporidium M. incurvata (Paskerova et al., 2016; Rotari, Paskerova & Sokolova, 2015; Sokolova et al., 2013)—in these articles, this gregarine is called Polyrhabdina sp.). Our transmission electron microscopic study of this eugregarine revealed peculiarities of the pellicle structure: the presence of loops of the internal lamina under epicytic grooves, which is not typical for eugregarines, and the absence of the internal lamina links at the epicytic crest bases, which is characteristic of some aseptate and septate eugregarines, e.g., Ancora and Stylocephalus (Desportes, 1969; Simdyanov, 1995; Simdyanov et al., 2017). Moreover, the AO of this gregarine is constructed as the epimerite of other studied eugregarines (see below). Based on the light, scanning and transmission electron microscopic data obtained in this study, we amend the diagnosis for P. pygospionis (see Taxonomic summary).

Eugregarines P. spionis and P. bifurcata (Mackinnon & Ray, 1931; Reichenow, 1932) were isolated from the polychaete Scolelepis fuliginosa (Claparède, 1868) (now Malacoceros fuliginosus). They were considered to be either variants of the same species P. spionis (Mackinnon & Ray, 1931) or separate species (Reichenow, 1932) distinguished by the number and morphology of prongs on the AO surface (Table S1). P. spionis has the epimerite with seven-nine bifurcated apical prongs, while P. bifurcata - with two large claw-like apical processes and a basal circlet of 14–16 min prongs. According to previously published drawings, the gregarines that we isolated from the polychaetes Malacoceros fuliginosus were more similar to P. spionis in appearance. Presumably, we observed young trophozoites of P. spionis judging from their reported cell sizes (Caullery & Mesnil, 1914a; Mackinnon & Ray, 1931; Reichenow, 1932; Table S3). Since we did not study the ultrastructure of Polyrhabdina cf. spionis, we cannot discuss the presence of prongs on the AO surface. Additionally, a collar at the AO base that we observed in P. pygospionis and P. cf. spionis has also been described in P. polydora (Caullery & Mesnil, 1914a; Kamm, 1922; Mackinnon & Ray, 1931; Fowell, 1936).

On the base of the superficial morphology of P. pygospionis and P. cf. spionis, the type species, and data on the fine structure of AO and cortex in P. pygospionis, we emend the diagnosis of the genus Polyrhabdina (see Taxonomy summary).

Kamm (1922) established the family Polyrhabdinidae for genera Polyrhabdina, Sycia Léger 1892, and Ulivina Mingazzini, 1891 uniting septate eugregarines with the epimerite-type AO. The species composition of these genera is still not defined because their morphological characteristics are insufficient and overlapping. The only species, S. inopinata Léger 1891, has been investigated by transmission electron microscopy. Sycia spp. and Ulivina spp. occur mostly in the Cirratulidae and Eunicidae polychaetes (Desportes & Schrével, 2013; Schrével, 1969; Schrével & Vivier, 1966).

Similar to Polyrhabdina spp., an area of light, non-granulated cytoplasm in the cell under the AO was revealed in Sycia and Ulivina eugregarines by light microscopic studies. However, neither our electron microscopic data on Polyrhabdina pygospionis, nor the data on the fine structure of S. inopinata confirm the presence of a fibrillar septum dividing the gregarine cell into the anterior and nucleated posterior parts, as characteristic of true septate eugregarines (Desportes & Schrével, 2013; Ganapati, 1946; Kamm, 1922; Mingazzini, 1891; Schrével, 1969; Schrével & Vivier, 1966; present study).

In gregarines of these genera, AOs vary from a small papilla with or without a long filament at the apex in Ulivina spp. to a large rounded papilla with a thick collar (“ring”) around the base in Sycia spp. The figures of S. inopinata in Schrével & Vivier (1966) and Schrével (1969), as well as personal communications of Prof. Schrével, allow suggesting that the collar is covered by the three-membrane pellicle and that the IMC terminates above the collar, but not under it as in P. pygospionis. Presumably, the collar of S. inopinata is a circular protrusion of the eugregarine cell under the globular AO. In addition, the granular cytoplasm with short fibrils in the AO base in P. pygospionis may correspond to a sphincter ring described in the AO in S. inopinata. In comparison with the pellicle of P. pygospionis, links of the internal lamina in the base of the epicytic crest as well as loops of the internal lamina under the bottom of epicytic grooves were apparently absent in S. inopinata (Desportes & Schrével, 2013; Schrével, 1969; Schrével & Vivier, 1966).

Due to scarce morphological and missing molecular data, the genera Sycia and Ulivina remain to be revised, and their relationships with Polyrhabdina need to be proved.

Epimerite is a shared characteristic of eugregarines

The fine structure of attachment organelles was investigated in archigregarines Selenidium spp., aseptate eugregarines Ancora sagittata, Difficilina cerebratuli, Lankesteria levinei, Lecudina spp., and septate gregarines Didymophyes gigantea, Epicavus araeoceri, Gregarina spp., Leidyana ephestiae, Pyxinia firmus, Stylocephalus africanus. A revision of these data (Simdyanov et al., 2017) revealed conspicuous differences between the attachment organelles of archigregarines and eugregarines and proposed to restrict the term “epimerite” to the AO in eugregarines, but the term “mucron” to the AO in archigregarines. The epimerite is an anchoring organelle, which varies in size and shape, and it is sometimes equipped with projections. It is usually embedded in the host cell invagination and bordered by a circular gap which runs around the AO base and pinches a small portion of the host cell. The epimerite is covered only by the plasma membrane, while the IMC of the pellicle terminates close to the circular gap at the epimerite base. Between the epimerite and infected host cell, a kind of tight cell junction is apparently formed. It may facilitate feeding by transmembrane transport. In trophozoites during their transformation from the zoite, the epimerite originates as a new organelle in front of their anterior end, with the simultaneous disappearance of some organelles of the apical complex characteristic of the zoite. As trophozoites develop into gamonts and detach from the host cell, they can lose the epimerite. In contrast, the mucron is usually small, rounded or sucker-shaped. It represents the hypertrophic developed anterior end of the zoite and performs feeding by using its well-developed apical complex as a cytostome-cytopharyngeal gateway for the myzocytosis - the feeding by sucking. The mucron persists in trophozoites and gamonts, and even at the syzygy stage. The recently obtained results on the fine structure of blastogregarines are in good agreement with the proposed hypothesis: blastogregarines, which share some plesiomorphic characters of archigregarines, possess the AO of similar structure and function of the mucron (Simdyanov et al., 2018).

According to our results, the AO of two Polyrhabdina spp. is organized in the same way as epimerites of other previously studied aseptate and septate eugregarines (Fig. 7). It represents an anchoring organelle covered only by the plasma membrane forming a kind of tight cell junction with the host cell membrane, and can be lost in gamonts. The evidence obtained in this study strongly corroborates the suggestion, that the epimerite is a shared, derived character (synapomorphy) of all eugregarines, both septate and aseptate, while the mucron is an intrinsic feature of archigregarines and blastogregarines—sporozoans preserving the apicomplexan zoite structure at the trophozoite and gamont stages (Simdyanov et al., 2017; Simdyanov et al., 2018). This revised view on AO homology is in agreement with the recently emended diagnosis of Eugregarinida (Simdyanov et al., 2017).

Figure 7 Diagram of the trophozoite structure of the eugregarine Polyrhabdina pygospionis attached to the host cell (not to scale).

Abbreviations: col, epimerite collar; cz, cortical zone of epimerite cytoplasm; ec, epicytic crests; ep, epimerite; f, fibrillar layer; gp, circular gap at the epimerite base under the collar; grm, granular material of the ectoplasma; h, host cell; hf, host cell circular fold; hm, host plasma membrane; il, internal lamina of the parasite pellicle; IMC, inner membrane complex of the parasite pellicle; N, nucleus; n, nucleolus; p, parasite cell; pm, parasite plasma membrane.

The resurrected family Polyrhabdinidae and the emended superfamily Ancoroidea

Molecular phylogenetic analyses of ribosomal RNA genes (SSU, 5.8S, and LSU rDNA) show that P. pygospionis typifies a large lineage of environmental eukaryotic sequences from marine sediments, which was previously shown to be related to Ancoroidea and designated as incertae sedis among gregarines (Simdyanov et al., 2017). Additionally, the recent protein multigene phylogeny also revealed the affiliation of P. pygospionis (Polyrhabdina sp. in original) with Ancoroidea (Janouškovec et al., 2019; Mathur, Wakeman & Keeling, 2021). We conclude that the clade including P. pygospionis can be identified as the family Polyrhabdinidae Kamm, 1922, a taxon that should be re-established with a diagnosis emended by new morphological evidence (see Taxonomic summary). The two subclades identified in our phylogenetic analysis in the clade Polyrhabdinidae may correspond to Polyrhabdina, Sycia and/or Ulivina genera, but this needs to be further investigated.

Geographical mapping of the polyrhabdinid environmental samples reveals their worldwide distribution: deep sea sediment from the East Sea (Park et al., 2008), sediment from seashores of Denmark (Karst et al., 2018), sediment of mangrove system in Brazil (Santos et al., 2010), methane cold seep in Sagami Bay in Japan, the tidal flat on Disko Island near Greenland, Cariaco basin in the Caribbean Sea near Venezuela, marine stromatolites near the Bahamas (Baumgartner et al., 2009; Stoeck, Taylor & Epstein, 2003; Stoeck et al., 2007; Takishita et al., 2007), and a clone derived from the foraminiferan Ammonia beccarii (Wray et al., 1995). The identification of the latter sequence as a foraminiferan was refuted (Pawlowski et al., 1996), and it probably originated by either pseudoparasitism, occasional ingestion of a gregarine oocysts by the foraminiferan (see Rueckert et al. (2011) for similar cases of misidentification).

SSU + LSU rDNA phylogeny, SSU rDNA phylogenies built with alternative alignment procedures, and the BI analysis performed in PhyloBayes confirm the placement of Trollidium akkeshiense within the Ancoroidea. In addition, most Trollidium, Polyrhabdinidae, and related environmental sequences share an unusual feature in the predicted secondary structure of helix 17 of the SSU rRNA that might indicate their common ancestry. Earlier, helix 17 was identified as a phylogenetic marker of some metazoan and protistan clades (Aleshin et al., 1998; Nikolaev et al., 2004). The clade associated with T. akkeshiense and several related environmental sequences was previously assigned to the family Lecudinidae (Rueckert & Leander, 2010; Rueckert, Wakeman & Leander, 2013; Wakeman, 2020). Following the criteria established in earlier taxonomic revisions of gregarines based on rDNA phylogenies (Clopton, 2009; Cavalier-Smith, 2014; Paskerova et al., 2018; Simdyanov et al., 2017), we erect the SSU rDNA clade, typified by T. akkeshiense, as a new family Trollidiidae (see Taxonomic summary). GUIDANCE2 alignment tends to place the Trichotokara+Paralecudina clade in the Ancoroidea, while the manually masked alignment brings these groups closer together only if using a BI under the CAT model (PhyloBayes tree), which is most free from the artifact of long branch attraction, or if excluding the most divergent sequences from the alignment (65 OUT tree). This is consistent with a recent multigene analysis of Trichotokara as a sister group to Ancoroidea (Mathur, Wakeman & Keeling, 2021). Generally, the GUIDANCE2 strategy showed less stable results than the structural alignment utilizing the X-INS-i algorithm, in particular with regard to placement of the Trichotokara and Paralecudina clade as well as the Cephaloidophora and Trollidiidae clades. The PCA result also showed that fully manual constructed alignments gave more stable results than the other two automatic alignments. The rDNA data offers method-dependent resolution for the composition of the Ancoroidea, and the results require further refinement. Currently, we propose that the superfamily Ancoroidea combines four families: Ancoridae Simdyanov et al., 2017, Polyplicariidae Cavalier-Smith, 2014, Polyrhabdinidae Kamm, 1922, emend., and Trollidiidae fam. nov. Characteristics of these groups are given in Taxonomic summary.

Candidate synapomorphies of the family Polyrhabdinidae and the emended superfamily Ancoroidea

Comparative analysis of the ultrastructural data and reconciling those with the molecular phylogenies allow us to detect additional candidate synapomorphies of Polyrhabdinidae and Ancoroidea.

In the attached eugregarines P. pygospionis and P. cf. spionis, almost the whole epimerite is embedded into the host cell, except for the collar, which is located above the surface of the host cell. The collar appears to be a shared characteristic of the genus Polyrhabdina and possibly also of the family Polyrhabdinidae, provided that the relationship of Sycia with this family is confirmed. The epimerite and its collar may provide a scaffold for formation of projections and prongs for better attachment to the host cell.

A shared characteristic among many gregarines that make up the clade Ancoroidea is a well-developed protruded epimerite detected in Ancora (family Ancoridae Simdyanov et al., 2017), Polyplicarium (Polyplicariidae Cavalier-Smith, 2014), Polyrhabdina (Polyrhabdinidae Kamm, 1922, emend.) (Cavalier-Smith, 2014; Cecconi, 1905; Simdyanov et al., 2017; Wakeman & Leander, 2013; this study). The epimerite is usually lost in parasites detached from the host cell. In place of the lost epimerite (discarded or retracted), a residual structure may be retained, e.g., a disk-shaped protrusion (“mucron” in original) observed in Trollidium akkeshiense (Wakeman, 2020).

Another shared characteristic and a candidate synapomorphy of the superfamily Ancoroidea is likely the absence of the links of the internal lamina in between the bases of the epicytic crests (Fig. 2I). Apart from P. pygospionis, the links are absent in Ancora sagittata (Simdyanov et al., 2017) and, re-examining pictures in the original studies, most likely also in Trollidium akkeshiense (Wakeman, 2020: Fig. 12), and in Sycia inopinata (Schrével, 1969: Fig. 23), a presumable member of the superfamily. Septate gregarines of the genus Stylocephalus (Desportes, 1969; Desportes & Schrével, 2013) also lack the links but are not related to Ancoroidea (Fig. 5), indicating that the two taxa acquired similar epicyte structure by convergent loss of the internal lamina links.

It is possible that the superfamily Ancoroidea includes other known species of marine gregarines. The eugregarines Kamptocephalus mobilis Simdyanov, 1995 and Mastigorhynchus bradae Simdyanov, 1995, parasitizing flabelligerid polychaetes, resemble T. akkeshiense by the fine structure of the epicyte and by the type of motility (Simdyanov, 1995; Wakeman, 2020). All these gregarines have wide epicytic crest without links of the internal lamina in their base and possess a well-developed epimerite, which is usually absent in gamonts. In addition, K. mobilis shows both intermittent bending of the anterior third of the body and fast gliding, and has a massive bundle of longitudinal microtubules in the subpellicular cytoplasm (Simdyanov, 1995). Hence, K. mobilis and M. bradae have all the features of Trollidium and ancoroids in general, but their affiliation is yet to be confirmed by molecular analyses.

Notes on gregarine co-parasitism and microsporidian hyperparasitism in polychaetes

Eugregarines Polyrhabdina species often co-occur with archigregarines Selenidium species in the same polychaete host (Table S3). Our observations suggest that in co-infection of P. pygospionis and S. pygospionis in Pygospio elegans (Paskerova et al., 2018; this study) each parasite species is more abundant than in monoinfections. Eugregarines, blastogregarines, and archigregarines regularly harbour metchnikovellid microsporidia (Caullery & Mesnil, 1897a; Caullery & Mesnil, 1897b; Caullery & Mesnil, 1919; Ganapati, 1946; Mackinnon & Ray, 1931; Mikhailov et al., 2021; Paskerova et al., 2016; Paskerova et al., 2018; Rotari, Paskerova & Sokolova, 2015; Sokolova et al., 2013; Sokolova et al., 2014; Table S3). Further studies focused on metchnikovellids that infect gregarines co-occurring in the same spionid polychaete may shed some light on the diversification of metchnikovellids and co-evolution of gregarine hosts and their hyperparasitic microsporidia.

Taxonomic summary

Phylum Apicomplexa Levine, 1970	
Subphylum Sporozoa Leuckart, 1879	
Class Gregarinomorpha Grassé, 1953, emend. Simdyanov et al., 2017	
Order Eugregarinida Léger, 1900, emend. Simdyanov et al., 2017.	
Superfamily AncoroideaSimdyanov et al., 2017, emend.	

Diagnosis. Eugregarinida. Typically aseptate; trophozoites typically with prominent epimerite; epimerite lost in gamonts; micropores on lateral walls of epicytic crests lacking the links of the internal lamina in their bases. In polychaetes, intestine.

Type family. Ancoridae Simdyanov et al., 2017.

Remarks. Four families. The superfamily may be supplemented with parasites of Flabelligeridae polychaetes, Kamptocephalus mobilisSimdyanov, 1995, Mastigorhynchus bradaeSimdyanov, 1995, which share the characteristics of the epicyte structure and motility with those of ancoroids (see Discussion). The affiliation of these gregarines with this superfamily needs to be tested.

Family Polyrhabdinidae Kamm, 1922, emend.

Diagnosis. Ancoroidea. Trophozoites and gamonts ovoid, aseptate; epimerite massive, with various appendages: prongs and/or basal collar (posteriorly oriented circular fold at the epimerite base). Intestine of polychaetes.

Type genus.PolyrhabdinaMingazzini, 1891.

Remarks. One to three genera. Sycia and Ulivina species have non-granulated cytoplasm in the cell body under the epimerite, which gives their appearance of septate cells. The validity and composition of these genera and its close relations to Polyrhabdina and other ancoroid eugregarines must be clarified.

Genus PolyrhabdinaMingazzini, 1891, emend.

Diagnosis. Polyrhabdinidae. Trophozoites and gamonts aseptate. Intestine of Spionidae.

Type species. Polyrhabdina spionis (Kölliker, 1848) Mingazinni, 1891.

Remarks. Seven named species.

Polyrhabdina pygospionis Caullery and Mesnil, 1914, emend.

Original description. Polyrhabdina pygospionis, n. sp., from Pygospio seticornis (now P. elegans) (Caullery & Mesnil, 1914a). Infected by microsporidia Metchnikovella incurvata Caullery et Mesnil, 1914 and M. oviformis Caullery et Mesnil, 1914 (Caullery & Mesnil, 1914b; Caullery & Mesnil, 1919). Gregarines generally abundant in the polychaete intestine, similar to P. brasili Caullery et Mesnil, 1914 from Spio martinensis Mesnil, 1896 but smaller (Caullery & Mesnil, 1919).

Re-description (amended diagnosis). Characteristics of the genus. Trophozoites and gamonts ellipsoid (sometimes slightly curved) to pear-shaped, circular in cross section, 28–288 × 14–50 µm. Nucleus oval to spherical, 9.5–19.0 × 9.5–17.0 µm, oriented longitudinally in the widest part of the cell, with single large or 2–3 small nucleoli of various localization. Epimerite domed, 3.7–4.9 µm in base diameter, with a posteriorly oriented circular fold (collar), 0.2–1.7 µm tall, and an annular narrow and deep gap at the base; lost (presumably discarded) in gamonts of different sizes. Epicytic crests (about 5/µm) with 10–12 apical rippled dense structures and 10-12 apical filaments in the tops. Other stages not found. Infected by different metchnikovellidean microsporidia.

Type locality. Anse Saint-Martin, English Channel, North East Atlantic.

Type definitive host.Pygospio elegans (former P. seticornis) Claparède, 1863 (Polychaeta, Spionidae).

Locality and host used in amended diagnosis. Kandalaksha Bay, White Sea; Pygospio elegans Claparède, 1863 (Polychaeta, Spionidae).

Ecology/Habitat. Marine.

Type materials. Lost.

Deposition of specimens and materials used inamended diagnosis. Resin blocks and fixed slides containing eugregarines and pieces of infected host intestine deposited in the collection of Department of Invertebrate Zoology, St Petersburg State University; Figs. 1–3 (this publication) show some of these specimens (White Sea). DNA sequences: contiguous sequence of SSU, ITS1, 5.8S, ITS2, and LSU rDNA from the individuals, isolated from the polychaetes Pygospio elegans (Kandalaksha Bay, White Sea) (GenBank accession number MT214481).

Family Trollidiidae fam. nov.

Diagnosis. Ancoroidea. Free individuals (putatively gamonts; epimerite unknown) with extraordinarily wide longitudinal epicytic crests, some crests zigzag; the network of longitudinal microtubules in cortex under the zigzag epicytic crests; bending/twitching motility. Intestine of Flabelligeridae. Monotypic.

Type genus. TrollidiumWakeman, 2020.

ZooBank Registration: LSID urn:lsid:zoobank.org:pub:693369E6-B319-4BB1-8E61-148FC4F5B271. ZooBank Nomenclature Act: LSID urn:lsid:zoobank.org:act:239658AC-6AE9-4641-B2F3-E18FE5616363.

Remarks. The family may be supplemented with other parasites of Flabelligeridae polychaetes, Kamptocephalus mobilisSimdyanov, 1995 and Mastigorhynchus bradae Simdyanov, 1995, which share with Trollidium the characteristics of the epicyte structure and motility (see Discussion). The affiliation of these gregarines with this family needs to be tested.

Conclusions

On the base of comprehensive study, we re-described the aseptate eugregarine Polyrhabdina pygospionis Caullery, Mesnil, 1914 from the polychaete Pygospio elegans, collected in the White Sea. We also demonstrated that the attachment organelles of P. pygospionis and P. cf. spionis, the type species, represented the epimerite in its organization. This evidence once again proves that the epimerite is an innovation of eugregarines. The phylogenetic analyses using concatenated ribosomal operon DNA sequences revealed that P. pygospionis was not related to lecudinoid eugregarines (the superfamily Lecudinoidea Simdyanov et al., 2017), but to ancoroid eugregarines (the superfamily Ancoroidea Simdyanov et al., 2017). Based on the results of ribosomal phylogenetic analysis and comparative analysis of the literature data, we revised the superfamily Ancoroidea and proposed the following synapomorphies for this group: the well-developed protruded epimerite usually missing in gregarines detached from the host cell and the absence of the links of the internal lamina joining the base of the epicytic crests. Accordingly, the superfamily unites four families: Ancoridae (Simdyanov et al., 2017), Polyplicariidae (Cavalier-Smith, 2014), Polyrhabdinidae Kamm, 1922, emend., and Trollidiidae fam. nov.

Supplemental Information

Supplemental Information 1 Gamonts of the eugregarine Polyrhabdina pygospionis detached from the host tissue. Light (A) and transmission electron (B) microscopy

All micrographs show gamonts infected with metchnikovellid microsporidia (Mi, presporogonial stage development). (A) Slightly compressed, detached gamont without the epimerite. Note the granular material (grm) in the ectoplasm of the anterior end and the nucleus (N). Differential interference contrast. (B) Oblique longitudinal section through the anterior end of a detached gamont without the epimerite. Note the granular material (grm) in the ectoplasm (ecto) of the anterior end, epicytic crests (ec), the internal lamina (il), and the nucleus (N).

Click here for additional data file.

Supplemental Information 2 Bayesian tree of eugregarines inferred from the manually masked dataset of 94 SSU rDNA sequences and 1,578 sites under the GTR+CAT+G8 model using PhyloBayes

Numbers at branches indicate Bayesian posterior probabilities. The newly obtained sequence of Polyrhabdina pygospionis is in bold.

Click here for additional data file.

Supplemental Information 3 Maximum likelihood trees

Maximum likelihood trees recovered from 1,578, 1,471, 1,366, 1,257, 1,126, and 828-site MAFFT E-INS-i + X-INS-i + trimAl and 1,574, 1,471, 1,366, 1,257, 1,126, and 828-site MAFFT E-INS-i + GUIDANCE2 alignments under GTR+F+I+G8 model with 1000 UFBoot replicates using IQ-TREE 2.1.2 (Minh et al., 2020). Numbers at branches indicate bootstrap (UFBoot) percentage supports.

Click here for additional data file.

Supplemental Information 4 Bayesian tree of eugregarines inferred from the manually masked dataset of 65 SSU rDNA sequences and 1,574 sites under the GTR+F+I+G8 model

Numbers at branches indicate Bayesian posterior probabilities (numerator) and ML bootstrap percentage (denominator). Black dots on the branches indicate Bayesian posterior probabilities and bootstrap percentages of 1.0 and 95% and higher, respectively. The newly obtained sequence of Polyrhabdina pygospionis is in bold.

Click here for additional data file.

Supplemental Information 5 R statistical computing

(A) Principal component analysis of alignments based on bipartition support values obtained in the ML analyses with UFBoot; twenty bipartitions with the most contributions to the principal components 1 and 2 are shown. Note that red dots formed a denser group than green or blues ones. Dot 828 indicates a critical level of data reduction at which the resolution of the trees is minimized. (B) Histogram of contribution values in percentages for the first ten bipartitions shown in A. (C–D) Comparison of two from the three alignments sets (Manual edited and GUIDANCE2; MAFT- X-INS-i is simalr to manual edited) in total tree length (C) and log likelihoods of trees (D); for both graphs, the maximum likelihood trees were used.

Click here for additional data file.

Supplemental Information 6 Alignment and secondary structure model for the helix 17 region in the 18S rRNAs of gregarines

Complementary nucleotides of the helices are shaded; the proposed evolutionary transition marked by a single nucleotide insertion and uniting the families Polyrhabdinidae and Trollidiidae is depicted schematically on the right; the corresponding scenario for the evolution of the helix 17 region within families Polyrhabdinidae and Trollidiidae is outlined in the tree (lower left) with at least 6 transitions: 1–deletion of 1 bp from the helix; 2 –expansion of the loop by 1 bp into the helix; 3 –1bp insertion in a single OTU; 4 –transformation of the inner loop into a bulge; 5 –one nucleotide deletion in the apical part of the 3′-strand (resulting in an internal loop –2 bp –bulge); 6 –transformation of the inner loop into a bulge (resulting in a bulge –2 bp –a bulge). The names of major eugregarine lineages correspond to (Simdyanov et al., 2017; Cavalier-Smith, 2014).

Click here for additional data file.

Supplemental Information 7 Motility of Polyrhabdina pygospionis. Light microscopy, differential interference contrast

Real-time video (15 s) of a detached trophozoite (the cytoplasm is flowing out of the cell in the place of the dislodged epimerite), slightly compressed with the coverslip and gliding forward.

Click here for additional data file.

Supplemental Information 8 Morphometry of investigated eugregarines

Abbreviations: av, average; SD, standard deviation; n, number of measurements.

Click here for additional data file.

Supplemental Information 9 Testing of possible compositions for the Ancoroidea

Anc –Ancoridae and Polyplicariidae, Poly –Polyrhabdinidae, Troll –Trollidiidae, Trich, Paralec –Trichotokara, Paralecudina and related environmental sequences, Ceph –Cephaloidiphoroidea, c-ELW –Expected Likelihood Weight (Strimmer and Rambaut 2002), p-AU –p-value of approximately unbiased (AU) test (Shimodaira, 2002). Plus signs denote the 95% confidence sets. Minus signs denote significant exclusion. All tests performed 10,000 resamplings using the RELL method in IQ-TREE 2.1.2 (Minh et al., 2020).

Click here for additional data file.

Supplemental Information 10 Diagnostic characteristics of eugregarines of the genus Polyrhabdina Mingazinni, 1891

Abbreviations: ‘—‘, no data; ‘?’, contradictory or vague description; ‘*’, species examined by electron scanning microscopy; ‘**’, species examined by electron scanning and transmission microscopy. The validation of the scientific names was conducted in the World Register of Marine Species (WoRMS).

Click here for additional data file.

Supplemental Information 11 Raw data of morphometry and infection rate of investigated eugregarines

Click here for additional data file.

Supplemental Information 12 The complete rRNA operon sequence (comprising the SSU, ITS1, 5.8S, ITS2, and LSU) of Polyrhabdina pygospionis available in GenBank: MT214481

Click here for additional data file.

The authors would like to thank the staff of the Marine Biological Station of St Petersburg State University, the White Sea Biological Station of Moscow State University for providing facilities for field sampling and material processing. GGP is grateful to Professor Rudolf Entzeroth and Markus Gunther (Technical University of Dresden) for providing facilities for her research. GGP and TSM utilized equipment of the core facility centers of St Petersburg State University: “Molecular and Cell Technologies” for electron microscopy and “Observatory of Environmental Safety” for culturing of marine invertebrates. AV and MK are grateful to the Laboratory of Electron Microscopy, Biology Centre CAS, an institution supported by the Czech-BioImaging large RI project (LM2015062 funded by MEYS CR), for their help with obtaining some EM data. TGS utilized equipment of the Electron Microscopy Laboratory of the Faculty of Biology and the Center of Microscopy of the White Sea Biological Station, Lomonosov Moscow State University. To make computations, the CIPRES Science Gateway (Miller, Pfeiffer & Schwartz, 2010) was used in this study. The research was carried out as part of the scientific project of the state order of the government of Russian Federation to Lomonosov Moscow State University No.121032300117-3.

Additional Information and Declarations

Competing Interests

Author Contributions

DNA Deposition

Data Availability

New Species Registration

The authors declare there are no competing interests.

Gita G. Paskerova conceived and designed the experiments, performed the experiments, analyzed the data, prepared figures and/or tables, authored or reviewed drafts of the paper, contributed to the material collection and processing, wrote the paper, and approved the final draft.

Tatiana S. Miroliubova, Andrea Valigurová and Vladimir V. Aleoshin conceived and designed the experiments, performed the experiments, analyzed the data, prepared figures and/or tables, authored or reviewed drafts of the paper, contributed to the material collection and processing, and approved the final draft.

Jan Janouškovec and Kirill V. Mikhailov conceived and designed the experiments, performed the experiments, analyzed the data, authored or reviewed drafts of the paper, contributed to the material collection and processing, and approved the final draft.

Magdaléna Kováčiková and Yuliya Ya. Sokolova performed the experiments, analyzed the data, prepared figures and/or tables, authored or reviewed drafts of the paper, contributed to the material collection and processing, and approved the final draft.

Andrei Diakin performed the experiments, prepared figures and/or tables, contributed to the material collection and processing, and approved the final draft.

Timur G. Simdyanov conceived and designed the experiments, performed the experiments, analyzed the data, prepared figures and/or tables, authored or reviewed drafts of the paper, contributed to the material collection and processing, wrote the paper, and approved the final draft.

The following information was supplied regarding the deposition of DNA sequences:

The complete rRNA operon sequence (comprising the SSU, ITS1, 5.8S, ITS2, and LSU) of Polyrhabdina pygospionis is available in the Supplementary File.

The following information was supplied regarding data availability:

All raw measurements and calculations are available in the Supplementary File.

Resin blocks and fixed slides containing eugregarines and pieces of infected host intestine deposited in the collection of the Department of Invertebrate Zoology, St Petersburg State University (accession numbers 3-6, 622-623, 678-680, 734, 786, 794-795, 862, 879 in the section “Polyrhabdina pygospionis”; accession numbers 742-743 in the section “Polyrhabdina cf. spionis”).

The following information was supplied regarding the registration of a newly described species:

Publication LSID: urn:lsid:zoobank.org:pub:693369E6-B319-4BB1-8E61-148FC4F5B271

Trollidiidae fam. nov.: urn:lsid:zoobank.org:act:239658AC-6AE9-4641-B2F3-E18FE5616363.

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
