# Peer review of "Evidence from the resurrected family Polyrhabdinidae Kamm, 1922 (Apicomplexa: Gregarinomorpha) supports the epimerite, an attachment organelle, as a major eugregarine innovation"

_PeerJ, doi:10.7717/peerj.11912_

## Round 0.1 · original submission · Major Revisions

Dear Dr. Paskerova and colleagues:

Thanks for submitting your manuscript to PeerJ. I have now received two independent reviews of your work, and as you will see, the reviewers raised some concerns about the research. Despite this, these reviewers are optimistic about your work and the potential impact it will have on research studying gregarine systematics. Thus, I encourage you to revise your manuscript, accordingly, taking into account all of the concerns raised by both reviewers.

Both reviewers agree that the manuscript is very well written, and the data are exceptional in presentation. The major problem seems to be a lack of strong support for the proposal of a new gregarine family. The morphological data are well-supported, yet the molecular evidence is not strongly compelling. It seems there is a misinterpretation of statistical support values, as there is a lack of support at several nodes, especially deep in estimated phylogenies. You might conduct further analyses to test the effect of masking on tree inferences, though additional analyses might be not helpful given minimal phylogenetic signal for resolving deeper nodes of estimated trees. This all creates concern with revising systematics and classification of the group at the family level.

There are other minor concerns raised by the reviewers. Please address all of these issues and consider making your taxonomic changes preliminary (or conditional) rather than formal, with a note of caution that the collective evidence is minimal at best.

I look forward to seeing your revision, and thanks again for submitting your work to PeerJ.

Good luck with your revision,

-joe

Reviewer 1 ·

Basic reporting

The study of Paskerova et al. is an important contribution to the morphological and molecular evolution of eukaryotic unicellular organisms from the phylum Apicomplexa. It is clearly and professionally written, authors have very good literature overview and expertise in studying apicomplexans from the class Gregarinomorpha. Figures are of excellent quality, well labelled and described. They are real eye-catchers.

Custom checks requested by Editor
- Authors provided DNA data deposition statement, including the GenBank accession number (MT214481). The data are, however, not available yet and will be released after publication according to the NCBI webpage.
- Authors did not describe any new species. They emended diagnoses of one species-group taxon (Polyrhabdina pygospionis) as well as of some higher taxa (i.e., Ancoroidea, Polyrhabdinidae, Polyrhabdina). Authors also established a new family, Trollidiidae. Its proposal meets standards of the ICZN (1999). Following Recommendation 8A of the International Commission on Zoological Nomenclature (2012), authors also provided the ZooBank registration number of the work. Although not obligatory, I also recommend to provide the ZooBank registration number of the act (establishment of the family), which was automatically generated when the new family-group name was registered.

Experimental design

The manuscript includes original primary research, with well defined questions. Morphological methods are state-of-the-art, they are well described. Molecular phylogenetic methods are standard and could be described in more detail and additional analyses should be conducted (please see General comments for the authors).

Validity of the findings

The characterization of Polyrhabdina pygospionis is of excellent quality. I am also convinced by the argumentation that epimerite represents the major morphological innovation of eugregarines. I also find the Taxonomic summary reliable, although statistical support for monophyly of the Ancoroidea is very poor (not strong as stated by the authors). Anyway, I do not see a problem in emendation of the diagnosis of the Ancoroidea, as this name is already available. Likewise, names of all other taxa treated in the 'Taxonomic summary' section are already available except for the Trollidiidae, which are established as a new family-group name. Trollidium obviously represents a very distinct evolutionary lineage within the Gregarinomorpha, so I do not see a problem to name its cluster.

Additional comments

Specific comments

- M & M - I would recommend to apply also a different alignment and masking strategy. For instance, authors can use the Mafft algorithm implemented in the Guidanace2 server and try different cut-off values to mask columns with certain confidence scores estimated by the alignment algorithm. It is much less subjective that to apply own masking strategy in BioEdit. At least five different cut-off values should be tried to see the robustness of results after building trees. Authors can compare these results with their own strategy and will see whether the aligning and masking strategy might have an impact on subsequent phylogenetic inferences.
It is not clear to me whether 5.8S rRNA gene sequences were treated as a separate partition or together with 28S rRNA gene sequences. 5.8S is rather short and has a very different evolutionary rate than 28S. Were partitions considered also in the ML analyses conducted with IQ-Tree? I find 100 bootstrap replicates to be very few. A minimum is 1000 bootstrap replicates to obtain reliable results. A rapid bootstrapping can be applied, but then different support thresholds apply.
I also find two phylogenetic methods to be insufficient, especially when the resolution in the deeper parts of trees is so poor. I do understand that this might be due to the lack of phylogenetic signal in the data, but I suggest to test this. There are also other ML (PhyML, RAxML) and Bayesian methods (PhyloBayes, Phycas) that could be employed for tree inferences.

- Results (Phylogenies inferred from rDNA sequences) - Authors incorrectly interpret statistical support values. Bootstrap values <70% are low, 70–94% are moderate, and ≥95% are high (Hillis, D. M. & Bull, J. J. 1993. An empirical test of bootstrapping as a method for assessing confidence in phylogenetic analysis. Syst. Biol., 42:182–192.). For the Bayesian posterior probabilities, values <0.94 are statistically insignificant and ≥0.95 are significant (Alfaro, M. E., Zoller, S. & Lutzoni, F. 2003. Bayes or bootstrap? A simulation study comparing the performance of Bayesian Markov Chain Monte Carlo sampling and bootstrapping in assessing phylogenetic confidence. Mol. Biol. Evol., 20:255–266.).

- Fig. 6 and 7 - please label archigregarines, blastogregarines etc.

- Diagnoses of taxa contain information (number of taxa included, monotypic etc.) that is not "diagnostic at all". Please remove it from there and transfer to remarks. Remarks to the superfamily Ancoroidea - it is not a well supported rDNA lineage (I see that there is post. probability 1.00, but 56% ML bootstrap. We cannot trust in such a lineage from a phylogenetic point of view).

Minor comments
line 54 - Ancoroidea Simdyanov et al., 2017 [please insert a comma, check this also in figure legends]
line 63 - Apicomplexa are [The Latin noun Apicomplexa is in plural, for instance Aves -> plural, we do not say "birds" is...]
line 310 - BP=56% is not moderate, it is very weak
line 320 - BP = 77% cannot be considered as strong, at best it can be considered as moderate
line 323 - posterior probability 0.94 is not high at all, it is statistically insignificant, BP <50% is nothing...
line 556 - Type family. Ancoridae Simdyanov et al., 2017 [please insert a comma]
line 563 - please delete "One to three genera." This adds nothing to 'Diagnosis' of the family.
lines 570-571 - please delete "Seven named species." This is not a diagnostic character. Maybe put this information to remarks and list all seven species.
line 609 - Please provide also ZooBank Registration of the act

·

Basic reporting

The paper is overall quite well written. I think it has many good qualities.

Experimental design

The paper has a good experimental design. It is a bit far-reaching with its conclusions in some places, based on the data presented, however.

Validity of the findings

Again, I think that the major flaw with this paper is that the conclusions, while well-stated, are far-reaching, and maybe need to be toned down a bit to better match the results. I think the authors are riding that line of speculation quite hard.

Additional comments

The manuscript entitled “Evidence from the resurrected family Polyrhabdinidae Kamm, 1922 (Apicomplexa: Gregarinomorpha) supports the epimerite, an attachment organelle, as a major eugregarine innovation” introduces new systematic organisations to gregarines, and also discusses some of the evolutionary history of the group, in particular the epimerite of eugregarines.

In general, I found the manuscript to be quite good. I do think it deserves to be published. The quality of the data is high, and I think the points brought up in the paper are interesting and are of academic significance to the field of gregarines which are poorly understood.

My major issue with the manuscript has to do with the introduction of new families, based on the molecular data presented in this paper. The trees generated from the ribosomal operon of gregarines are just not resolved at the appropriate nodes to use this information to support changes to systematics and classification of the group at the family level (in my opinion). Looking at the trees, support for Ancoroidea (including Trollidium) is 1.0/56 and the next node down is 0.98/32. It is just not great support. I know that the reduced-taxon tree shows 0.94/90, and 1.0/75, but this could also be a relic of long branch attraction. These branches are some of the longest on the whole tree of eukaryotes. I propose that perhaps we can get more taxa in these larger, multigene datasets and make more informed decisions on the families and genera at that time, rather than changing things repeatedly and just causing increased confusion in the field. One of the section headings in the paper reads “Ribosomal DNA supports the family Polyrhabdinidae and the emended superfamily Ancoroidea.” I just don’t think the data supports it that strongly.

Nonetheless, I think that the paper has some interesting points, and the data really is of high quality, so I think it should be published. I just want to avoid having superfamilies introduced every time we publish a new gene. It could get quite tedious. But I will leave that to the editor to make the final decision.

Best,

Kevin Wakeman

---

## Round 0.2 · accepted · Accept

Dear Dr. Paskerova and colleagues:

Thanks for revising your manuscript based on the concerns raised by the reviewers. I now believe that your manuscript is suitable for publication. Congratulations! I look forward to seeing this work in print, and I anticipate it being an important resource for groups studying gregarine systematics. Thanks again for choosing PeerJ to publish such important work.

Best,

-joe

Reviewer 1 ·

Basic reporting

The study of Paskerova et al. is an important contribution to the morphological and molecular evolution of eukaryotic unicellular organisms from the phylum Apicomplexa. It is clearly and professionally written, authors have very good literature overview and expertise in studying apicomplexans from the class Gregarinomorpha.

Experimental design

The manuscript includes original primary research, with well defined questions. Morphological methods are state-of-the-art, they are well described. Molecular phylogenetic analyses were very carefully conducted, the effect of various alignment and masking strategies on tree inferences was adequately addressed.

Validity of the findings

The characterization of Polyrhabdina pygospionis is of excellent quality. The argumentation that epimerite represents the major morphological innovation of eugregarines is very convincing. Taxonomic summary is consistent and well prepared.

Additional comments

Dear Authors,

I am highly satisfied with the revision! All my concerns were appropriately addressed. I am also very happy to see that phylogenetic analyses were moved to superior level. In my opinion, the present multifaceted phylogenetic approach added a lot to the future strategies of tree inferences in Apicomplexa and can serve as inspiration for other researchers.

Congratulations!